

# A 28-year-long (1997–2024) hydrographic dataset from the southern Baltic Sea

Daniel Rak[1], Anna Izabela Bulczak[1], Waldemar Walczowski[1], Piotr Wieczorek[1], Małgorzata Merchel[1], Robert Osiński[1], Ilona Goszczko[1], Agnieszka Beszczynska-Moeller[1], Agnieszka Strzelewicz[1], Małgorzata Kitowska[1]

[1] Observational Oceanography Laboratory, Institute of Oceanology PAN, Physical Oceanography Department, Sopot, Poland

* Correspondence to: Daniel Rak (rak@iopan.pl)

**Abstract.** The data set presented here consists of Conductivity–Temperature–Depth (CTD) observations collected during 96 research cruises of R/V *Oceania* across the southern Baltic Sea between 1997 and 2024. The collection comprises towed and vertical station profiles acquired along a repeat transect spanning the Arkona Basin, Bornholm Basin, Słupsk Furrow, and Gdańsk Basin. Acquisition and post-processing procedures include standardized parsing of CNV/TXT files, robust time/position handling, pressure-binning to 1 dbar, median filtering, automated geolocation quality control, and pruning of incomplete profiles. The dataset enables analyses of seasonal to decadal variability in temperature and salinity, inflow propagation, ventilation events, and model validation. Manufacturer specifications for the principal instruments (Guildline 87104, Idronaut OS316/OS316Plus, Sea-Bird SBE49, Sea-Bird SBE19plus) are summarized to inform uncertainty assessment.

## 1. Introduction

The Southern Baltic Sea, a semi-enclosed marginal sea, creates an important transitional area connecting the North Sea through the Danish Straits. This unique geographic location makes distinct hydrographic conditions characterized by significant inflows of saline waters from the North Sea, meaningfully influencing local marine dynamics (Matthäus & Franck, 1992; Mohrholz et al., 2015). Consequently, this region exhibits pronounced vertical stratification patterns driven by both inflow events and its relative distance from the Danish Straits, coupled with distinct seasonal variability in temperature and salinity (Leppäranta & Myrberg, 2009).

Major Baltic Inflows (MBIs), occur episodically from the North Sea into the Baltic Sea, significantly affecting its hydrography and circulation (Fischer & Matthäus, 1996; Mohrholz et al., 2015). These inflows are classified into two main types: barotropic and baroclinic. Barotropic inflows result primarily from large-scale meteorological forcing, such as prolonged westerly winds, significant changes in atmospheric pressure, and sea level differences between the North Sea and Baltic Sea, causing substantial volumes of saline water to enter the Baltic basins (Burchard et al., 2005, Stigebrandt and Gustafsson, 2003). Baroclinic inflows, on the other hand, are driven by density gradients and stratification differences, often involving internal waves and subsurface transport mechanisms. MBIs transport large volumes of dense, oxygen-rich saline waters into deeper Baltic basins,

replenishing oxygen levels in bottom waters and impacting both physical and ecological processes (Mohrholz,
2018). The frequency, intensity, and impact of these inflow events are crucial for understanding the long-term
environmental status of the Baltic Sea, influencing deep-water renewal and ecosystem dynamics (Reissmann et
al., 2009).
Comprehensive observational datasets for the Southern Baltic Sea remain relatively scarce, particularly
continuous, high-resolution Conductivity-Temperature-Depth (CTD) profiles spanning multiple decades (Feistel
et al., 2008). While sporadic measurement campaigns and shorter-term datasets exist, long-term, systematic
collections are limited, making it challenging to fully understand the variability and long-term trends in the region
(Omstedt et al., 2014). This scarcity is especially pronounced within the Polish Exclusive Economic Zone (EEZ),
where the availability of openly accessible, high-resolution CTD data is particularly limited. In contrast to the
better-monitored central basins of the Baltic Sea—such as the Bornholm and Gotland Basins—data coverage in
the Polish EEZ has been historically sparse and fragmented. Regular measurements in this area have often been
conducted only a few times per year, and real-time or near-real-time data have not been readily available until the
recent deployment of Argo floats (Walczowski et al., 2020). The lack of dense, long-term in situ CTD records
hinders detailed analyses of vertical structure, stratification, oxygen dynamics, and long-term hydrographic shifts
in this environmentally and economically important sector of the Baltic Sea. Furthermore, this data gap poses
significant challenges to numerical modeling, operational oceanography, and marine environmental management.
Addressing this critical gap, this article introduces a unique, meticulously curated dataset comprising CTD profiles
collected over 28 years (1997–2024) in the Southern Baltic Sea by the Observational Oceanography Laboratory
of the Institute of Oceanology Polish Academy of Sciences (IOPAN), Physical Oceanography Department. The
data, gathered systematically aboard the research vessel R/V Oceania, provide invaluable insights into the physical
oceanographic processes shaping the local environment and their connection to broader climatic phenomena. Some
of these data have already been used in previous publications on long-term changes in the southern Baltic Sea,
including studies on temperature and salinity (Rak and Wieczorek, 2012), oxygen levels (Rak et al., 2020), the
upper ocean mixing and stratification (Bulczak et al., 2024) and the sea energy (Rak et al., 2024), and inflow
propagation (Rak, 2016). However, in this work, we make the full dataset publicly available and provide a
comprehensive description of its processing and key features.
The significance of these measurements lies in their extensive temporal coverage and high spatial resolution, which
enable comprehensive analyses of variability across a range of temporal scales—from seasonal to decadal—and
spatial scales, from sub-mesoscale to basin-wide. Furthermore, this dataset serves as a vital resource for improving
numerical ocean modeling and validation efforts, contributing significantly to our understanding of both local
marine dynamics and global climate-related processes (Gröger et al., 2021).

## 2. Study area and campaign design

The repeat hydrographic section follows the axis of the deep basins in the southern Baltic Sea, from the Arkona
Basin through the Bornholm Basin and the Słupsk Furrow to the Gdańsk Basin. This section spans approximately
280 nautical miles (≈ 519 km), and the measurement time with the towed probe varies from 3 to 5 days, depending
on weather conditions.



The transect evolved over time. Since the early 2000s, the core transect from the Bornholm Channel
(Bornholmsgat) to the Gdańsk Basin has been performed in a highly repeatable manner. In contrast, the
westernmost segment across the Arkona Basin was not fixed: some transects terminated closer to the Øresund,
whereas others extended toward the Darss Sill, depending on logistics and weather.
In the years preceding 2020, budget cuts reduced available sea time and progressively curtailed the western reach
of the section—first to the vicinity of the Bornholm Channel and ultimately to profiles within the Polish Exclusive
Economic Zone, which remains the current operational limit.
Over the 28-year period (Figure 1), sampling effort concentrated on hydraulic controls and mixing "hot spots" that
govern the pathway and transformation of North Sea inflow waters. Consequently, data density is the highest in
the Bornholm Channel and across the Słupsk Sill and its eastern flank, with sustained coverage along the main
axis toward the Gdańsk Basin.


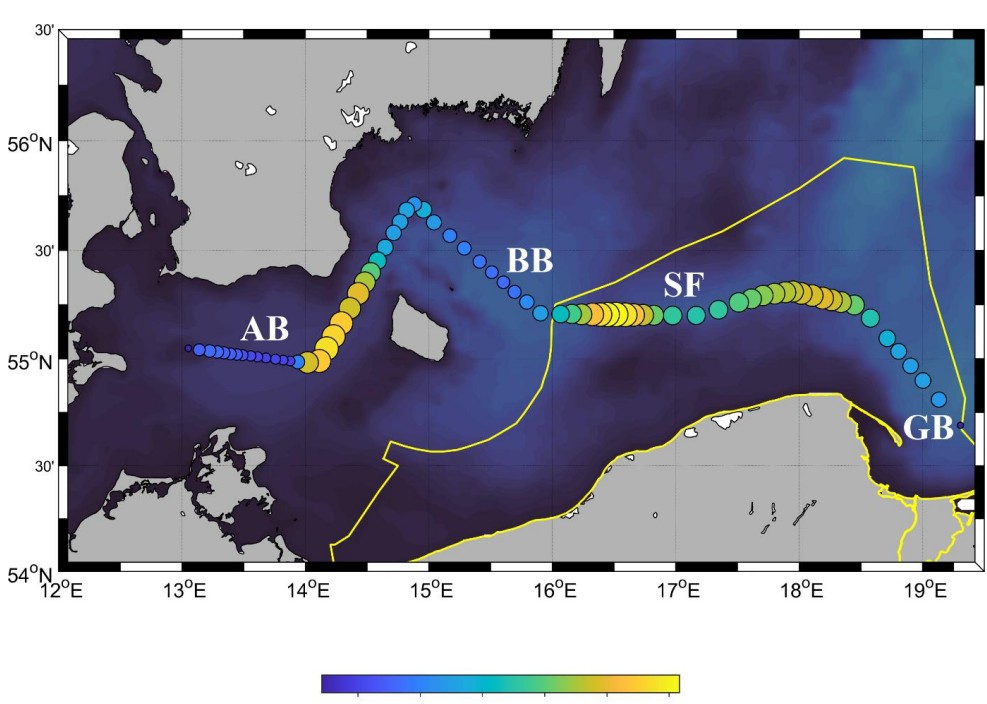


**Figure 1: Spatial distribution of CTD profiles collected during R/V Oceania cruises between 1997 and 2024 in the**
**southern Baltic Sea. The yellow line indicates the Polish Exclusive Economic Zone (EEZ). The labels AB, BB, SF and**
**GB denote the Arkona Basin, Bornholm Basin, Slupsk Furrow and Gdańsk Basin, respectively.**



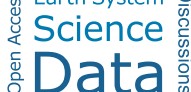

### 3. Instruments and measurement modes

Hydrographic observations at IOPAN were conducted with several CTD systems (Table 1). Early operations used a Guildline 87104 and an Idronaut OS316; the OS316 was soon complemented and largely superseded by the Sea-Bird SBE49 FastCAT. All of these instruments were initially deployed in towed mode (underway profiling). Since 2020, vertical casts (stations) profiling has largely replaced towing, primarily with a Sea-Bird SBE19plus and, more recently, an Idronaut OS316Plus.

**Table 1: Overview of CTD instruments used by IOPAN for hydrographic observations in the southern Baltic Sea between 1997 and 2024. The table lists the main devices and corresponding measurement types conducted during each period.**

| Year | CTD system | Measurement type |
|------|------------|------------------|
| **1997–1999** | Guildline 87104 | Towed |
| **2000–2002** | Idronaut OS316 | Towed |
| **2002–2003** | Idronaut OS316; Sea-Bird SBE49 | Towed |
| **2004–2020** | Sea-Bird SBE49; SBE19plus | Towed + vertical casts |
| **2021–2023** | Sea-Bird SBE19plus | vertical casts |
| **2023-2024** | Idronaut OS316Plus / Sea-Bird SBE19plus | Towed + vertical casts |

During towing, the CTD is mounted in a protective metal frame with an under-slung chain to minimize the risk of seabed contact. This configuration maintains a stable, near-horizontal probe orientation while providing mechanical protection (Figure 2). To produce a near-sinusoidal sampling pattern, the probe is cycled repeatedly between surface and bottom. At a towing speed of ~4 kn, this yields a horizontal resolution of ~200–500 m in typical water depths of 60–120 m. Towed data are acquired on both the downcast and upcast. Since 2020, vertical stations with a nominal along-track spacing ~5 nm have replaced towing. Owing to the probe's mounting and its orientation relative to the direction of motion, only the downcast is retained for vertical (station) measurements.

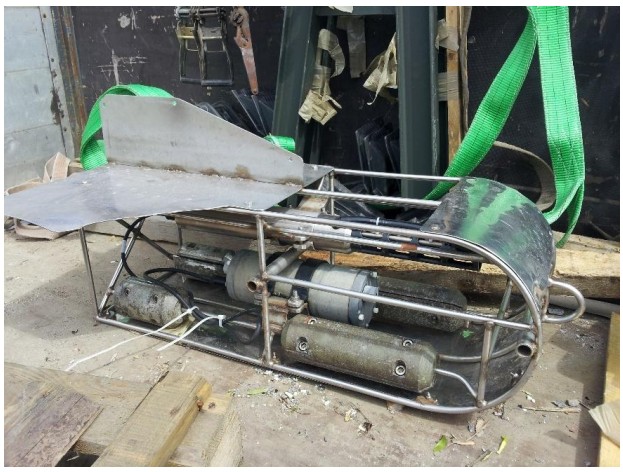


**Figure 2: CTD towed probe system used for the collection of data with Sea-Bird SBE49 (2002-2020)**


Instrument choice for the towed platform was driven by high sample-rate capability and robust real-time telemetry.
The Sea-Bird SBE19plus has been the primary shipboard profiling CTD on board R/V *Oceania*, whereas the
Idronaut OS316Plus—initially used at stations—has more recently been integrated into a refurbished towed frame.
A further advantage of the OS316Plus is its pass-through interface that allows additional auxiliary sensors (e.g.,
dissolved oxygen, turbidity) to be powered and telemetered over a single cable. Manufacturer accuracy
specifications for all instruments used in this program are summarized in Table 2.
**Table 2: Manufacturer specifications of conductivity-temperature-depth (CTD) profilers used by the Institute of**
**Oceanology, Polish Academy of Sciences (IOPAN) during long-term hydrographic monitoring in the southern Baltic**
**Sea.**

| CTD system | Pressure Accuracy | Temperature Accuracy | Conductivity Accuracy | Sampling | Notes |
|---|---|---|---|---|---|
| **Guildline 87104** | (no public spec) | (no public spec) | (no public spec) | (no public spec) | Manufacturer specs not found publicly for this model. |
| **Idronaut OS316S** | ±0.05 % of full-scale (0-7000 dbar) | ±0.003 °C | ±0.003 mS/cm in salt water | 4–16 Hz | Full-ocean-depth probe, pump-free, seven-ring quartz conductivity cell. |
| **Idronaut OS316Plus** | 0.05 % of full scale (standard); 0.01 % FS | ±0.002 °C | ±0.003 mS/cm | 12–20 Hz (real-time), typically 20 Hz | Higher precision version, pump-free, 1500 dbar white POM housing. |
| **Sea-Bird SBE49 FastCAT** | ±0.1 % of full-scale | ±0.002 °C | ±0.0003 S/m (≈ ±0.003 mS/cm) | 16 Hz | Autonomous CTD for vehicles/ROVs; high sample rate. |
| **Sea-Bird SBE19plus / V2** | ~±0.1 % of full scale (strain- | ~±0.005 °C | ~±0.0005 S/m (≈ ±0.005 mS/cm) | 4–6 Hz | Widely used profiling CTD, version V2 adds |



| gauge) / ~±0.02 % FS (quartz) | quartz option for improved pressure accuracy. |
|---|---|



### 4. Dataset and methods

Observational data utilized in this study were gathered during research voyages of the Institute of Oceanology of
the Polish Academy of Sciences' vessel, R/V Oceania. These data originated from the Southern Baltic Sea,
spanning the period between 1997 and 2024 (Figure 1). Approximately four transects were conducted annually,
with an aim to evenly cover the entire year. Due to the engagement of R/V Oceania in Arctic research, the period
from June to August is the least represented in this dataset.

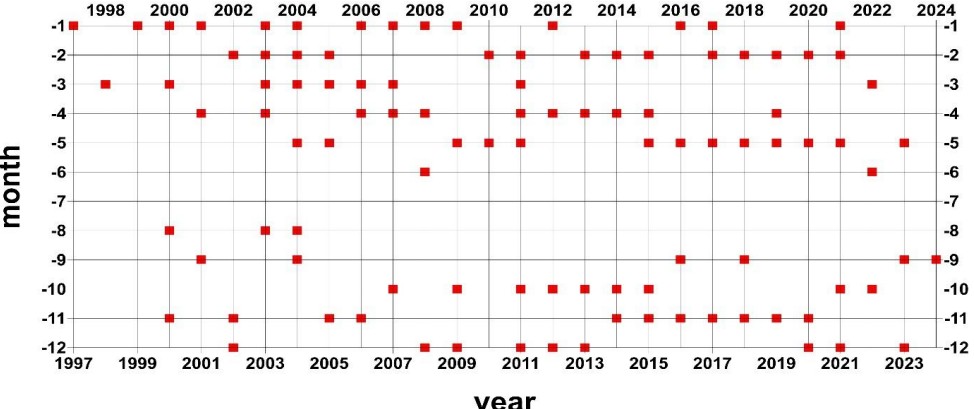


**Figure 3: Time distribution (1997–2024) of R/V Oceania CTD cruises conducted by the Observational Oceanography**
**Laboratory, Institute of Oceanology PAN.**
In total, from 1997 to 2024, 96 hydrographic voyages were conducted, during which 55032 measurement profiles
were recorded (Figure 4). The annual profile counts show strong interannual variability with a clear maximum in
the early–mid 2000s, when intensive towed CTD surveys routinely yielded several hundred to >1,000 profiles per
year. From the late 2000s into the 2010s the effort gradually declined, reflecting reduced sea time and a growing
share of discrete station work. A sharp drop is evident after 2019, consistent with the loss of the SBE49 towed
system in May 2020 and COVID-19 operational constraints; only sparse profiles were collected in 2020–2024.
Overall, the variability primarily reflects instrument availability, cruise logistics, and weather, rather than changes
in processing or quality control.

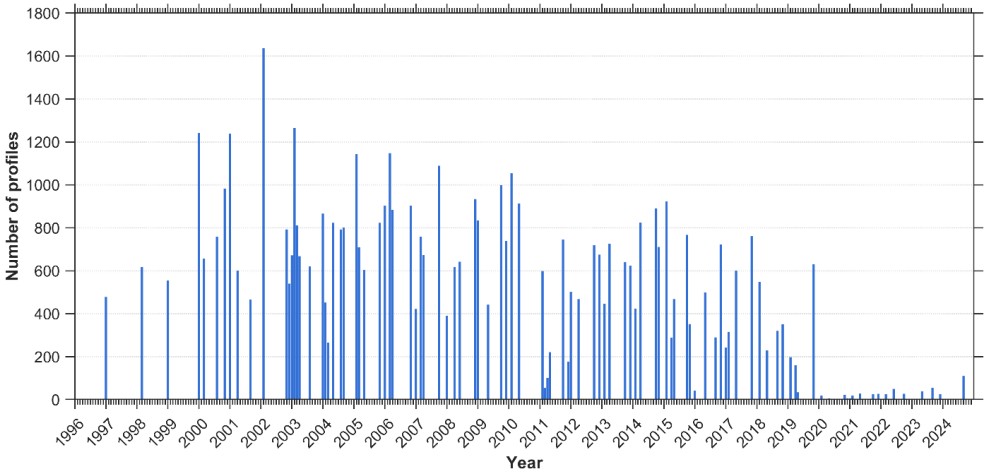

**Figure 4: Distribution of CTD profiles conducted by IOPAN along the main monitoring transect in the southern Baltic Sea.**

Towed measurements made with Guildline 87104, Idronaut OS316, and Sea-Bird SBE49 store a single geographic position and timestamp at the beginning of each profile. As the probe trails behind the vessel, the horizontal position error increases with depth. A simple geometric argument assuming an extreme cable angle yields an upper bound of about three times the local depth (e.g. at 100 m depth, near-bottom data could in principle be offset by up to ~300 m), but in practice actual offsets are substantially smaller (see Section Instrument calibration and uncertainty budget).

For vertical (station) profiles, this error is much smaller and arises primarily from the ship's drift during the cast. With the Idronaut OS316Plus, each sample is associated with a geographic position; however, this is the ship's GPS position, so the tow-induced offset still applies.

As illustrated in Figure 5, the descent rate is not uniform: it varies when crossing the pycnocline and with irregular ship motion. Additionally, operators deliberately slow the winch near the surface and close to the seabed for safety, further modulating sampling speed.

From 1997 to 2020, towed operations were routine. On 20 May 2020 the towed system carrying the Sea-Bird SBE49 was lost; since then, measurements have been conducted predominantly as vertical station casts, with limited towing resumed in 2023–2024 using a refurbished frame. In the post-2020 period, towing is applied selectively due to weather constraints and reduced ship time (typically ~7-day cruises vs ~14 days in earlier intensive periods, e.g., around 2003), and is therefore combined with a reduced set of station casts.



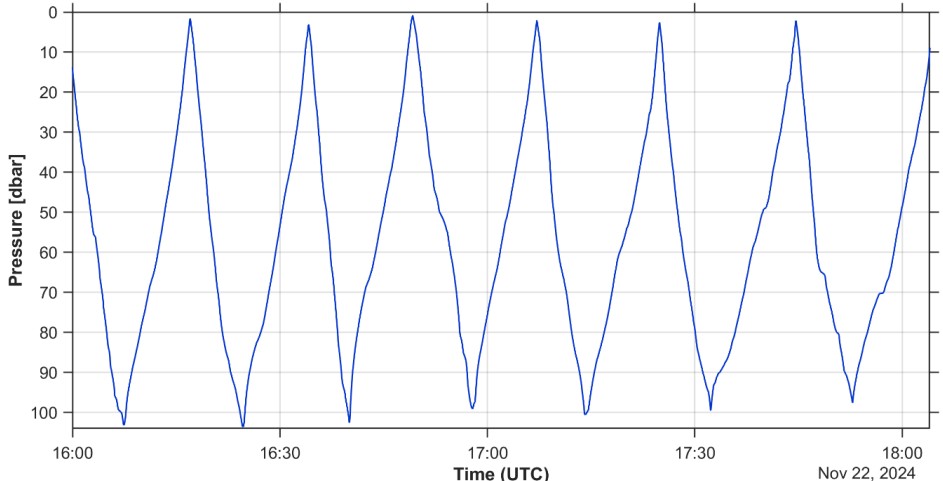


**Figure 5: Example of a typical continuous CTD probe trajectory in the water column during towed measurements.**


### 4.1 Instrument calibration and uncertainty budget

All Sea-Bird sensors used in this dataset (SBE 49 FastCAT, Sea-Bird SBE19plus / V2) were regularly returned to
Sea-Bird GmbH (Kempten, Germany) for servicing and post-cruise calibration, typically every 1–3 years,
depending on instrument usage and cruise schedules. Service reports document routine post-cruise calibration of
temperature and conductivity sensors, calibration of pressure sensors, firmware updates and full system checks for
the SBE 49 FastCAT and associated pump and conductivity modules.
For the most frequently used CTD on the towed system (Sea-Bird SBE 49 FastCAT), the latest post-cruise
calibration performed in March 2018 yielded extremely small residuals relative to laboratory standards.
Temperature calibration over the range 1–32.5 °C showed residuals within ±0.0001 °C, i.e. more than an order of
magnitude smaller than the nominal manufacturer accuracy.
Conductivity calibration residuals were on the order of $10^{-4}$ S m$^{-1}$ across the full range of bath salinities, i.e.
negligible compared to the nominal conductivity accuracy.
Pressure calibration for the 870-psia (≈ 600 dbar) pressure sensor showed residuals within ±0.01 % of full scale,
effectively at the limit of the calibration procedure.
Based on manufacturer specifications and these post-cruise calibration results, we adopt conservative instrumental
uncertainties of ±0.005 °C for temperature, ±0.01 in practical salinity and ±0.5 dbar for pressure for individual 1-
dbar binned measurements from Sea-Bird CTDs. These values are larger than the formal calibration residuals, but
they account for potential long-term sensor drift between service intervals and any residual biases introduced by
data processing (vertical binning, median filtering) and deployment configuration (e.g. slight lags due to pump
response and flow through the conductivity cell). For other CTD models used earlier in the time series (Guildline
87104, Idronaut OS316), we adopt comparable or slightly larger uncertainties consistent with manufacturer
specifications and our internal cross-comparisons (Table 3). The Idronaut OS316Plus was factory-calibrated at
Idronaut (24 Nov 2025). Although the dataset analysed here ends in 2024, we report the most recent factory
calibration to document instrument performance and to motivate the conservative uncertainties adopted for the




processed products. Calibration residuals were within ±0.0011 °C for temperature and ±0.0039 mS/cm for
conductivity. For consistency with the long-term record and to account for drift between service intervals and
processing effects, we adopt conservative uncertainties of ±0.005 °C, ±0.01 PSU, and ±1 dbar for 1-dbar binned
products.

**Table 3: Overview of the main CTD instruments used in the dataset and the conservative instrumental uncertainties**
**adopted for temperature, practical salinity (PSS-78; reported here in PSU) and pressure.**

| Period | Main CTD model | Calibration interval | Temperature uncertainty (°C) | Salinity uncertainty (PSU) | Pressure uncertainty (dbar) |
|---|---|---|---|---|---|
| **1997–1999** | Guildline 87104 | No data | No data | No data | No data |
| **2000–2003** | Idronaut OS316 | every 3 years | ±0.01 | ±0.02 | ±1 |
| **2004–2020** | SBE 49 / SBE19plus | every 1—2 years | ±0.005 | ±0.01 | ±0.5 |
| **2023—2024** | Idronaut OS316Plus | every 2 years | ±0.005 | ±0.01 | ±1 |


In addition to instrumental uncertainties, there is a finite spatial representativeness error associated with towed
sections. During towing, the CTD is pulled astern and may therefore be horizontally displaced from the ship's GPS
position. Simple geometric considerations show that an extreme upper bound of three times the local depth would
require the tow cable to be almost horizontal, which is unrealistic for our operating conditions (towing speeds of
≈ 4 kn and depths of 60–100 m). In practice, observed cable angles correspond to horizontal offsets of the order
of 0.3–0.8 times the local depth; for the error budget we therefore adopt the local depth as a conservative upper
limit on horizontal position uncertainty (≈ 100 m at 100 m depth), while typical offsets are likely closer to 0.5
times the depth. We therefore recommend that the dataset be used primarily for basin-scale and mesoscale
analyses, rather than for resolving fine-scale (< O(100 m)) frontal structures, where unresolved horizontal offsets
may become non-negligible.
**5.  Quality check and postprocessing of CTD data**
The quality control (QC) and postprocessing procedures applied to the CTD data collected by IOPAN are essential
for ensuring the scientific value, internal consistency, and long-term usability of the dataset. Raw data were
recorded using several CTD systems operated over the multi-decade period and were calibrated according to
manufacturer recommendations.
Postprocessing starts with an automated MATLAB routine that imports CNV/TXT files and parses station
metadata (date/time and geographic coordinates). Raw samples are first screened for common acquisition artefacts:
records with negative pressure (p < 0 dbar) are removed, and instrument fill values (e.g., $-9.990\times10^{-29}$) are
converted to NaN. Each cast is then sorted by pressure, repeated pressure levels are consolidated by averaging,



and the profile is standardized onto a uniform vertical axis (0–199 dbar, $\Delta p = 1$ dbar) using local averaging within
a $\pm 1$ dbar window around each target level. Temperature and salinity are denoised using a running median filter
(movmedian, window size typically 20 samples, omitting NaNs). Profiles with missing or invalid metadata are
excluded by masking casts where longitude/latitude/time are equal to 0 or NaN. For the gridded processing stream,
navigation is additionally quality-checked using broad domain limits (lon 13–22°E, lat 54.2–58°N) and a despiking
rule that flags positions deviating by more than 0.1° from a 5-point running median; flagged lon/lat/time values
are propagated as a common mask across variables.
Hydrographic variables are further checked for physically implausible values and unstable segments. Salinity is
constrained to a plausible range (7.2–21), with values outside this range set to NaN. Profiles are truncated below
the first occurrence of $\geq 5$ consecutive NaNs in salinity, and casts exhibiting an abrupt salinity decrease larger
than 0.01 between adjacent levels are terminated from that depth downward. A static-stability check based on
TEOS-10 density (computed with gsw_rho) is applied for $p < 200$ dbar; levels that would imply a density inversion
were flagged by setting the affected salinity values to NaN. For the gridded product, an additional optional vertical
smoothing step is applied using a moving mean (smoothdata, window 10) to reduce residual small-scale noise
while retaining vertical gradients.
After automated QC and postprocessing, profiles are aggregated into structured arrays, enabling downstream
climatological and statistical analyses. The processed data are routinely visualized to identify potential outliers
and systematic artefacts, and a manual review complements the automated steps, particularly for casts affected by
strong ship motion or transient sensor behaviour. To illustrate the effect of our pipeline, Figure 6 contrasts a raw
cast with its post-processed counterpart for cruise B0515. This cast was selected as a stress-test case with
pronounced motion/sensor transients near the halocline. Residual small-scale noise visible in the raw data is only
lightly attenuated by design: our post-processing is intentionally conservative to preserve mesoscale gradients and
avoid over-smoothing that could bias stratification metrics.



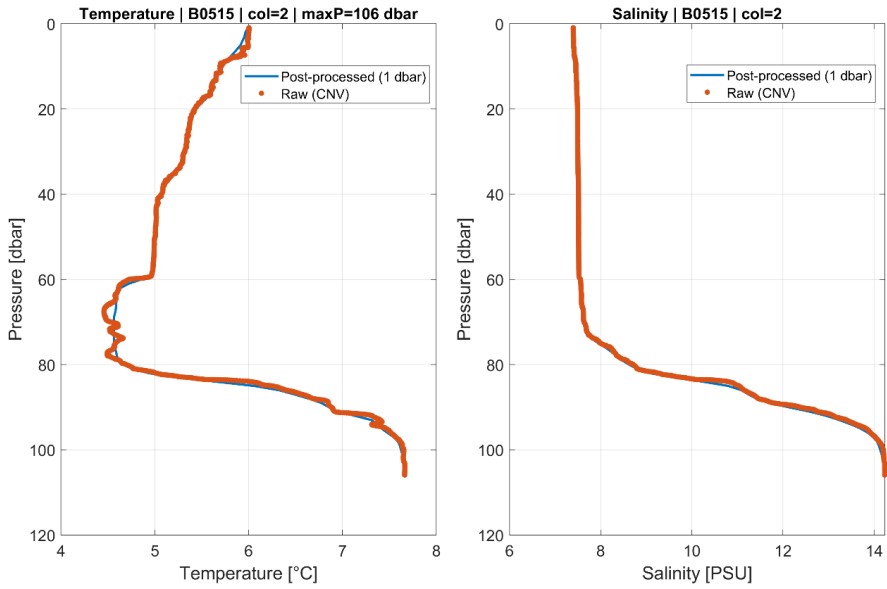


**Figure 6: Example of a raw CTD profile and the same profile after post-processing. Raw CNV samples (dots) and the 1-dbar product (solid line) are shown for temperature (°C) and salinity (PSU); pressure increases downward (dbar).**



### 6.  Data structure and export

The dataset is delivered in two interoperable formats. First, as a single MATLAB container in which each cruise
is stored as a separate field of the top-level struct IOPAN. Cruise fields follow the BMMYY convention (B –
Baltic; MM – month; YY – year; e.g., B0523 for May 2023) and contain gridded, column-oriented hydrographic
profile matrices together with a shared vertical coordinate. Second, the same cruise-wise products are exported as
a collection of per-cruise NetCDF files (IOPAN_BMMYY.nc) compliant with the CF-1.8 conventions and the
discrete sampling geometry (DSG) profile representation, with depth × profile hydrographic variables and profile-
wise (1-D) geolocation and time coordinates.

MATLAB (IOPAN_Baltic.mat)
For each cruise:
• Pressure, Temperature, Salinity: size N×M, where rows are 1-dbar levels and columns are individual profiles.
• Pressure_string: size N×1, the common vertical grid (e.g., 0:1:199 dbar).
• Time (MATLAB datenum), Longitude, Latitude: stored as size N×M for convenience and co-registration with
the hydrographic matrices; within each profile (column) these values are constant with depth (i.e., they represent
station-level metadata repeated along the vertical). Timestamps represent MATLAB serial days (fractional part =
time of day) and should be interpreted as UTC unless stated otherwise.

NetCDF (IOPAN_BMMYY.nc)





The NetCDF files use a compact CF-DSG layout in which:
• Hydrographic variables are stored on a common pressure grid (dbar) as 2-D arrays with dimensions (pressure,
profile), with 1-D coordinates lon(profile), lat(profile), and time(profile).
• Coordinates lon(profile) and lat(profile) are stored as 1-D profile-wise variables.
• Time is provided as a 1-D CF-compliant coordinate time(profile) (e.g., seconds since 1970-01-01 00:00:00 UTC,
calendar = gregorian).
During export, the station-level time and position are obtained from the MATLAB N×M matrices by extracting a
representative value per profile (e.g., the first finite value in each column).
A metadata block accompanies the cruise fields and documents units, creation timestamp, ownership and contact
point. Missing or filtered values are encoded as NaN in MATLAB and as _FillValue in NetCDF; pruning during
processing removes empty profiles (all-NaN columns) and ensures consistent dimensions within each cruise. The
processing and export workflow is implemented in the MATLAB scripts build_IOPAN_from_CNV_TXT.m and
write_IOPAN_to_netcdf.m (Zenodo, https://doi.org/10.5281/zenodo.17814769).

### 7. Basin-scale hydrographic structure and variability

Basin-scale vertical structure along the repeat transect is summarized in Figure 7. In all four basins, the upper ~40–
60 dbar are dominated by seasonally varying, relatively fresh surface waters, as reflected in the broad temperature
envelope and modest salinity range. Below the seasonal thermocline, temperature variability is much reduced,
while salinity exhibits a pronounced step-like increase associated with the permanent halocline. The mean
halocline depth and deep salinity systematically change along the section: in the shallow Arkona Basin
stratification is comparatively weak and confined to the upper water column, whereas the Bornholm Basin and
Słupsk Furrow display strong, well-defined haloclines overlying more saline deep waters. Toward the Gdańsk
Basin, deep salinities decrease and the halocline shoals slightly, consistent with progressive dilution and mixing
of North Sea inflow waters along their downstream pathway. The shaded ranges highlight that, despite substantial
interannual and event-scale variability, the basic vertical structure and along-transect contrasts between basins are
robust features of the 28-year record.

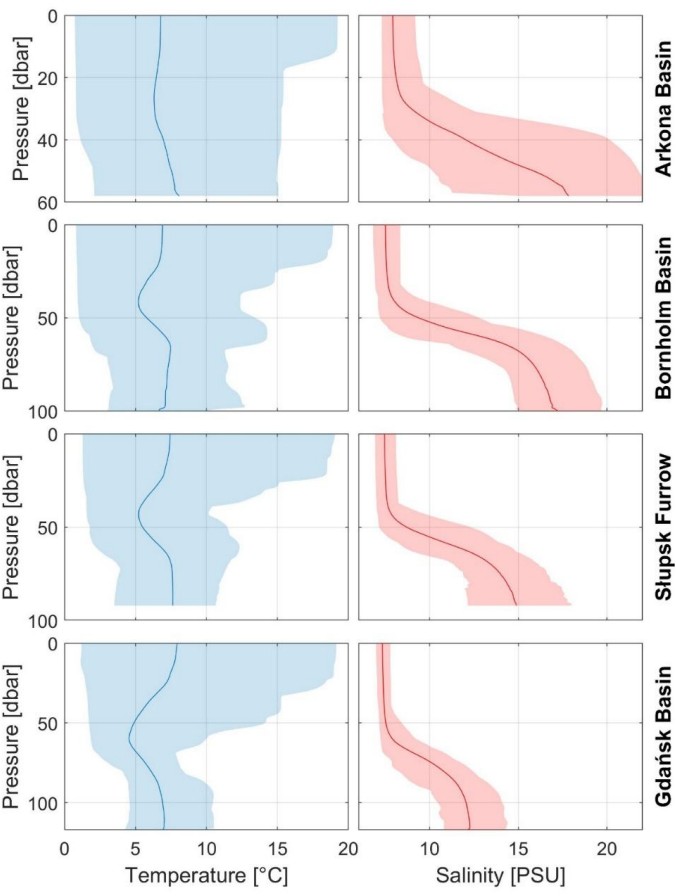


**Figure 7: Basin-mean vertical profiles of temperature and salinity in the Arkona Basin, Bornholm Basin, Słupsk Furrow**
**and Gdańsk Basin derived from all CTD casts collected along the repeat section between 1997 and 2024. Solid lines**
**indicate the multi-year mean and shaded envelopes the full range (min–max) across all cruises.**


Monthly mean temperature sections (January–December) along the repeat Baltic transect are shown in Figure 8.
The 12-panel climatology highlights the pronounced seasonal cycle of the upper water column, with winter cooling
and a deep, relatively homogeneous mixed layer followed by spring onset of stratification, summer surface
warming and development of a shallow thermocline, and an autumnal erosion of stratification. A persistent
dichothermal (cold intermediate) layer is visible from approximately April through November, reflecting winter-
cooled water retained below the seasonal thermocline while the surface layer warms. Along-transect differences
reflect the changing basin geometry and hydrographic regime from the Arkona Basin through the Bornholm Basin
and Słupsk Furrow toward the Gdańsk Basin, and the month-to-month variability in deeper layers suggests that
advection plays a key role in shaping the subsurface temperature field along the monitoring section.



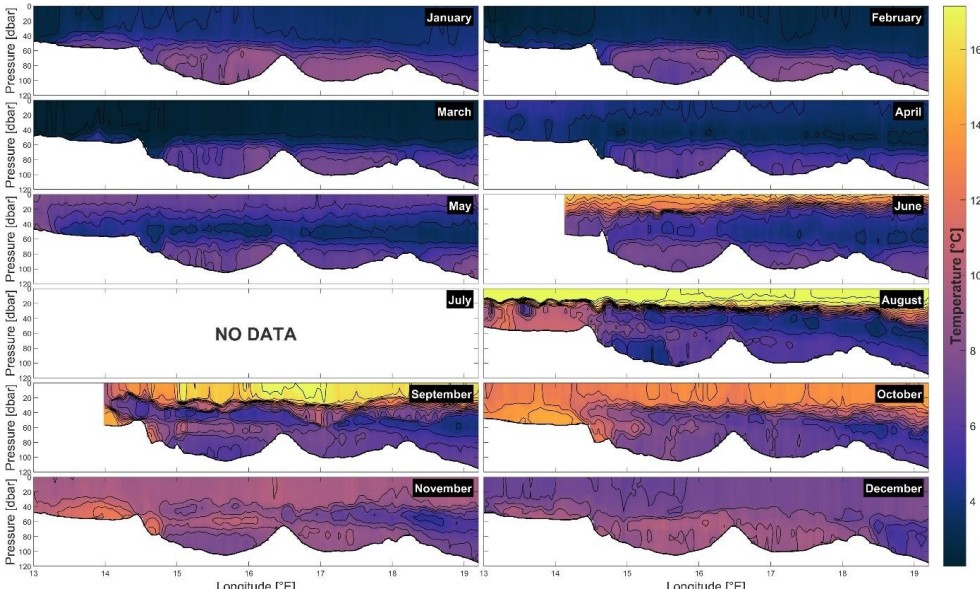

**Figure 8: Monthly mean temperature sections (January–December) along the Baltic transect as a function of longitude (°E) and pressure (dbar).**

Figure 9 shows the multi-year mean salinity section along the repeat southern Baltic transect, plotted as a function of longitude (°E) and pressure (dbar). The upper ~40–60 dbar is dominated by relatively fresh surface waters, while a distinct step-like increase in salinity below marks the permanent halocline; this halocline is most pronounced over the Bornholm Basin and Słupsk Furrow, which also host the highest salinities in deeper layers. Farther east toward the Gdańsk Basin, deep salinity decreases and the halocline structure changes along the section, reflecting the along-transect hydrographic contrasts and downstream modification of more saline waters.

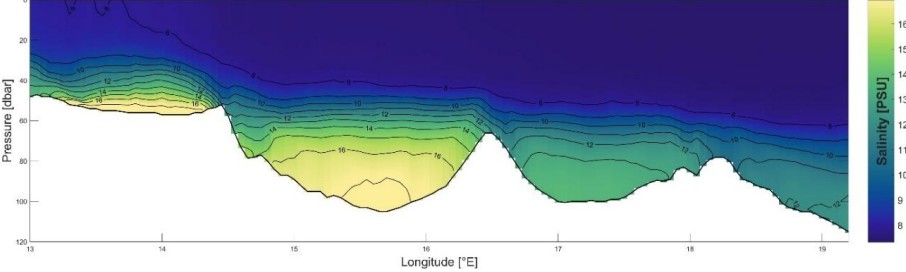

**Figure 9: Mean salinity (1997—2024) section along the repeat Baltic transect, shown as a function of longitude (°E) and pressure (dbar).**

Complementary basin-mean time series of layer-averaged temperature and salinity (Figure 10) illustrate how this vertical structure evolves in time. The 0–10 dbar surface layer is characterized by large interannual variability in temperature and relatively modest changes in salinity, reflecting the combined influence of atmospheric forcing, riverine input, and local mixing. In contrast, the bottom layer (defined consistently as the interval from 20 dbar



above the seabed down to the bottom) varies more episodically, with pronounced salinity and temperature
anomalies associated with inflow-driven ventilation events and subsequent stagnation periods. Along the transect,
these deep-layer signals are strongest in the Bornholm Basin and Słupsk Furrow, and become progressively
attenuated toward the Gdańsk Basin, in line with the downstream transformation of dense inflow waters inferred
from the vertical structure in Figure 7.

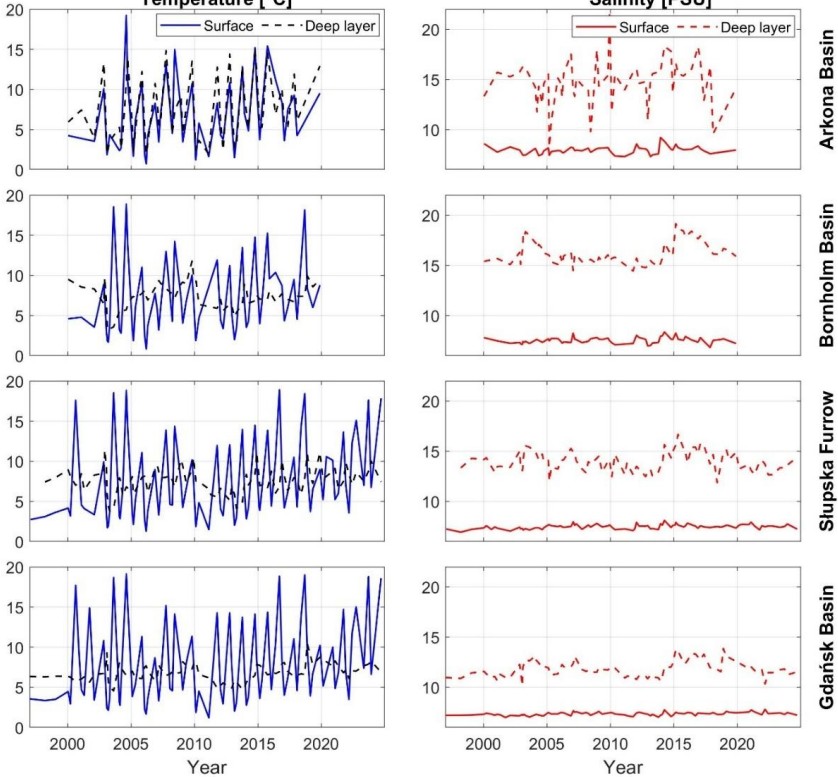


**Figure 10: Basin-mean time series of layer-averaged temperature (left) and salinity (right) in the Arkona, Bornholm, Słupsk Furrow and Gdańsk basins for 1997–2024. Solid lines show the surface layer (0–10 dbar), while dashed lines show a bottom layer defined uniformly from 20 dbar above the bottom down to the bottom in each basin.**






## 8.  Conclusion

We present a unique, quality-controlled CTD dataset spanning 1997–2024, assembled along a repeat section from the Arkona Basin through the Bornholm Basin and Słupsk Furrow to the Gdańsk Basin. In total, 96 cruises and 55,032 profiles were collected, providing rare temporal continuity and along-track resolution for the southern Baltic Sea.

The observing system evolved from high-rate towed profiling to a hybrid approach that, since 2020, also includes vertical station casts with a nominal $\leq 5$ nm spacing. Instrumentation progressed from Guildline 87104 and Idronaut OS316 to Sea-Bird SBE49 and SBE19plus, and most recently Idronaut OS316Plus—choices driven by sampling-rate capability and robust telemetry. Together, these modes and sensors yield horizontal scales of ~200–500 m at ~4 kn in 60–120 m depths and enable both down- and up-cast sampling in tow.

A consistent processing chain—standardized parsing of CNV/TXT, robust time/position handling, binning to 1 dbar, median filtering, automated geolocation QC, and pruning of incomplete casts—ensures inter-comparability through time and across instruments. The final distribution package (IOPAN_Baltic.mat) provides cruise-wise fields (BMMYY) with 1-dbar vertical grids, co-registered P–T–S matrices, and MATLAB serial-day time stamps (UTC), ready for analysis and conversion.

We explicitly acknowledge limitations that inform interpretation: reduced summer coverage due to Arctic commitments; tow-induced horizontal uncertainty comparable to the local depth (with typical offsets $\approx 0.5\times$ depth; see Section Instrument calibration and uncertainty budget); and a marked post-2020 decline in sampling linked to the loss of the SBE49 towed system (May 2020) and COVID-19 constraints. These variations predominantly reflect instrument availability, logistics, and weather rather than changes in QC or processing.

Despite these constraints, the dataset fills a long-standing observational gap in the Polish EEZ, where long, high-resolution CTD time series have been scarce, thereby strengthening model validation, reanalysis, and process studies of stratification, mixing, and inflow-driven ventilation along hydraulic controls. The section also serves as a reference line for quality control and validation of Baltic Argo float profiles and other autonomous observations, anchoring their measurements in a well-characterized hydrographic framework.

By making the full resource publicly available in both MATLAB and CF-compliant NetCDF formats, with transparent methods and structure, we provide an immediate foundation for multi-scale studies—from seasonal to decadal variability in temperature and salinity to the propagation and transformation of North Sea inflows—and for data assimilation in regional models. Continued observations along this established transect, ideally with renewed high-rate towed capability and routine auxiliary sensors (e.g., dissolved oxygen, turbidity) leveraging the OS316Plus pass-through interface, will be essential for tracking ongoing hydrographic change and supporting evidence-based management in the Baltic Sea.





**Code availability**

The MATLAB scripts used to build the processed IOPAN structure (Rak, 2025) from raw exports and to generate the CF-1.8 NetCDF products (build_IOPAN_from_CNV_TXT.m and write_IOPAN_to_netcdf.m) are archived on Zenodo at https://doi.org/10.5281/zenodo.17814769.

**Data availability**

The full dataset (Rak, 2025) is available from https://doi.org/10.48457/IOPAN.2025.531 in two formats: a single MATLAB file (IOPAN_Baltic.mat) containing all cruises as the IOPAN struct, and a collection of per-cruise, CF-1.8-compliant NetCDF files (IOPAN_BMMYY.nc, one file per cruise). Both formats provide the same gridded hydrographic fields and associated metadata, enabling straightforward use in MATLAB, Python and other common analysis environments.

**Author contribution**

DR coordinated the compilation of the IOPAN CTD dataset, designed the processing and quality-control workflow, processed and quality-controlled the data, developed the MATLAB processing and NetCDF export scripts, and prepared the figures and the initial manuscript draft. All authors contributed to CTD data acquisition during the cruises, participated in discussions on data interpretation and quality assessment, reviewed the manuscript, and approved the final version.

**Competing interests**

The authors declare that they have no conflict of interest.

**Acknowledgements**

We gratefully acknowledge the late Prof. dr hab. Jan Piechura, co-author of this study, for his invaluable contribution to the collection of the hydrographic data used here. We also thank the crew of r/v Oceania for their long-term support during the field campaigns.

**Funding**

This publication was supported by the following project: Argo-Poland, funded by the Polish Minister of Education and Science [grant number 2022/WK/04].





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
