# Peer review of "A 28-year-long (1997–2024) hydrographic dataset from the southern Baltic Sea"

_Earth System Science Data, 2025_

## Referee Comment (RC2)

**A 28-year-long (1997–2024) hydrographic dataset from the 1 southern Baltic Sea**
**By Rak Daniel et al.**

**Reviewer Comments:**

This manuscript presents a long-term CTD dataset from the southern Baltic Sea spanning the period 1997–2024, based on 96 research cruises that include both towed and vertical station profiles acquired along a repeat transect covering the Arkona Basin, Bornholm Basin, Słupsk Furrow, and Gdańsk Basin. In total, 55,032 measurement profiles were collected over the 28-year period. The manuscript provides a clear description of the instruments used and the quality-control and post-processing procedures applied. It also includes a basic climatological and statistical analysis of the dataset.

The manuscript is generally well written, with good English grammar, well-structured and significant, and it describes the quality-control procedures and data processing in a clear manner. However, my main comment is that, although the manuscript is not intended as a scientific analysis paper, several observed features are described only superficially and in rather vague terms, without fully exploiting the scientific potential of this unique long-term dataset.

Therefore, I recommend mayor revision

**Minor comments:**

**Hereafters are my comments section by section**

**1. Introduction**

The designation of the study area as *unique* appears ambitious based on the current description. It would be more appropriate either to better substantiate this claim by highlighting specific distinguishing characteristics of the area or to remove the term altogether.

In addition, the overview of previous measurements in the region is incomplete. It is unclear whether observations have been conducted in the past by other institutions or countries. Although references are provided, the introduction should more clearly convey the extent to which the area has been sampled historically, while placing greater emphasis on the contribution of the IOPAN measurements, which currently appear to be discussed mainly in relation to the most recent campaigns.

Furthermore, for readers who are not familiar with the region, the introduction should better explain the role of the proximity to the Danish Straits and how it influences the stratification and hydrographic characteristics of the area. Additional historical evidence of circulation patterns and variability in the area should be included.

**2. Study area and campaign design**

The description of the transects and sections is not sufficiently clear. It is not explicitly stated whether the transects were fully completed or which specific sections were carried out during the campaign. While this information can be inferred from the number of casts, providing a rough estimate (e.g. in percentage) of the completed transects and sections would significantly improve clarity for the reader.

Figure 1 should be enhanced by including bathymetric characteristics and clearly indicating the sampling sites, as well as key features such as the Darss Sill, Bornholm Channel, Słupsk Sill, and Øresund.

**3. Instruments and measurement modes**

The rationale for the chosen spatial resolution, specifically the sampling of vertical stations every 5 nautical miles, is not clearly explained and should be justified. Additionally, it is unclear whether discrete salinity samples were collected for calibration or validation purposes; if not, this should be explicitly stated and discussed.

**4. Dataset and methods**

In **line 127**, it is stated that four transects were conducted annually. However, it is not clear whether this refers to four individual transect segments (i.e. AB, AB–BB, BB–SF, and SF–GB) or to four repetitions of the entire transect. This distinction should be clarified, as providing more detailed information on the sampling strategy would substantially strengthen the methodological description and add value to the study.

**Figure 3** would benefit from the inclusion of an additional line below month 12 summarizing the total number of campaigns, which would help the reader better interpret the temporal coverage of the dataset.

**At line 154**, station profiles are typically associated with a fixed geographic position. For short-duration casts, the positional variability of the vessel is generally limited unless strong currents are present in the study area. If significant currents affected station positioning, this should be explicitly mentioned and discussed. It is also unclear whether the absence of NMEA data transmission applies exclusively to the Idronaut OS316 Plus system or whether this information was provided for another reason; clarification is needed.

**At line 157**, the use of the term *"deliberately"* is misleading and should be replaced with wording that reflects standard operational practice. A rephrasing is recommended.

> **4.1 Instrument calibration and uncertainty budget:** line 178 value or conductivity calibration residuals should be in mS/cm as units in table 2, otherwise provide Idronaut units in S/m there

Finally, the implications of the reduced sampling frequency after 2000 on the climatological and statistical analyses are not sufficiently addressed. The authors should discuss how this change affects data representativeness, uncertainty, and the robustness of the derived statistics.

**5. Quality check and post-processing of CTD data**

**At line 227**, the origin of the applied salinity range is not specified. It should be clarified whether this range is derived from visual inspection of the data or from a regional climatology and, if so, which dataset was used. In addition, it would be useful to indicate whether the climatology incorporates observations from other platforms, such as Argo floats. Have any of the CTD profiles been compared with available Argo profiles in the region? Such a comparison would strengthen confidence in the quality-control procedures and the overall robustness of the dataset.

**6. Data structure and export**

Considerable investment has been made by the European Union in initiatives such as EMODnet and SeaDataNet, as well as within the Copernicus framework, to promote FAIR data principles and facilitate data sharing within the scientific community. It is therefore important to clarify whether the presented dataset is accessible through any additional repositories or databases beyond those described in the manuscript.

If the data are not shared through established European infrastructures, the reasons for this choice should be explicitly stated. Furthermore, given the recent Copernicus initiatives aimed at rescuing and harmonizing historical datasets, it would be valuable to discuss whether these data could be integrated into such efforts.

**7. Basin-scale hydrographic structure and variability**

The presentation of basin-scale hydrographic structure and variability would benefit from a clearer link between the observed patterns and the underlying physical processes. In particular, the discussion could be strengthened by explicitly relating the observed data results to known circulation features, water mass exchanges, and stratification dynamics in the basin (as you stated in your introduction).

**Figure 7** – I suggest using the same scale for all the graphs (i.e., also for the Arkona Basin).
**Figure 8** – The figure clearly shows nicely the seasonal evolution of temperature; however, there is no information provided about salinity. Even if you do not include a graph, you should at least provide a brief description. Does the inflow of salinity exhibit seasonal variability as the temperature?

***Figure 9*** – I don't really get the message why you have a mean salinity graph for the whole period. In general Bornholm Basin and Słupsk Furrow host the highest salinities in deeper layers (lines 316-317 and line 329), so as seen also in *figure 10*. It is worth commenting this.

Figure 10 – What is the rationale behind for not presenting a mean time series for the intermediate layer? This layer seems to be very dynamic and worth to be presented.